# Axillary Surgery for Breast Cancer in 2024

**DOI:** 10.3390/cancers16091623

**Published:** 2024-04-23

**Authors:** Martin Heidinger, Walter P. Weber

**Affiliations:** 1Breast Surgery, University Hospital Basel, 4031 Basel, Switzerland; martin.heidinger@usb.ch; 2Faculty of Medicine, University of Basel, 4001 Basel, Switzerland

**Keywords:** breast cancer, axillary surgery, tailored axillary surgery, sentinel lymph node biopsy, axillary lymph node dissection

## Abstract

**Simple Summary:**

Historically, all patients with breast cancer (BC) underwent radical removal of lymph nodes under the armpit and up to the neck. Since the 1990s, axillary surgery has become increasingly de-escalated, and few indications for axillary lymph node dissection (ALND) remain. Patients with small BC (<2 cm) and unremarkable clinical examination through palpation and ultrasound may safely forego any axillary surgery. For patients with clinically node-negative BC and up to two positive lymph nodes found on sentinel lymph node biopsy, ALND can be safely avoided. If no residual tumor cells are found in the lymph nodes after neoadjuvant chemotherapy (NACT), ALND is not necessary. Ongoing studies are investigating whether axillary radiotherapy can provide similar survival outcomes to ALND in patients with clinically node-positive BC or in patients with residual nodal disease after NACT.

**Abstract:**

Axillary surgery for patients with breast cancer (BC) in 2024 is becoming increasingly specific, moving away from the previous ‘one size fits all’ radical approach. The goal is to spare morbidity whilst maintaining oncologic safety. In the upfront surgery setting, a first landmark randomized controlled trial (RCT) on the omission of any surgical axillary staging in patients with unremarkable clinical examination and axillary ultrasound showed non-inferiority to sentinel lymph node (SLN) biopsy (SLNB). The study population consisted of 87.8% postmenopausal patients with estrogen receptor-positive, human epidermal growth factor receptor 2-negative BC. Patients with clinically node-negative breast cancer and up to two positive SLNs can safely be spared axillary dissection (ALND) even in the context of mastectomy or extranodal extension. In patients enrolled in the TAXIS trial, adjuvant systemic treatment was shown to be similar with or without ALND despite the loss of staging information. After neoadjuvant chemotherapy (NACT), targeted lymph node removal with or without SLNB showed a lower false-negative rate to determine nodal pathological complete response (pCR) compared to SLNB alone. However, oncologic outcomes do not appear to differ in patients with nodal pCR determined by either one of the two concepts, according to a recently published global, retrospective, real-world study. Real-world studies generally have a lower level of evidence than RCTs, but they are feasible quickly and with a large sample size. Another global real-world study provides evidence that even patients with residual isolated tumor cells can be safely spared from ALND. In general, few indications for ALND remain. Three randomized controlled trials are ongoing for patients with clinically node-positive BC in the upfront surgery setting and residual disease after NACT. Pending the results of these trials, ALND remains indicated in these patients.

## 1. Introduction

Axillary surgery for breast cancer (BC) has evolved significantly from a previous “one size fits all” approach that involved radical surgery, including lymph node dissection extending from the axilla to the neck, to an increasingly granular and individualized surgical treatment. Axillary lymph node dissection (ALND) was the standard of care for all patients with BC until the nineties, which was considered to be a therapeutic procedure. The rationale behind it was that a complete surgical removal of locoregional tumor residues would result in improved survival, a hypothesis that has never been proven and was already questioned by the NSABP-04 trial [1,2,3]. In this landmark study, patients with clinically node-negative and node-positive BC were shown to have similar 10-year overall survival outcomes, no matter whether ALND or axillary radiotherapy (ART) were performed [3]. These results could be confirmed in clinically node-negative patients, who underwent breast-conserving surgery (BCS) and adjuvant radiotherapy of the breast [4]. Therefore, radical surgery and its associated morbidity was increasingly questioned. Axillary staging information was still deemed necessary, leading to the development of the sentinel lymph node (SLN) biopsy (SLNB). Whilst showing a false-negative rate of around 10%, excellent oncologic outcomes were achieved [5,6,7,8]. Notably, SLNB dramatically reduced surgical morbidity and improved quality of life [5,9,10,11,12,13,14,15,16,17]. Nevertheless, approximately 5% of patients still experience surgery-related morbidity [9]. Therefore, studies to identify patients in whom surgical axillary staging can be altogether abandoned have been initiated.

In the present manuscript, current evidence on axillary surgery for BC in the upfront surgical setting, after neoadjuvant chemotherapy (NACT) and in special situations such as inflammatory BC and locoregional recurrence, are reviewed.

## 2. Axillary Surgery in the Upfront Surgery Setting

After having become the standard of care in clinically node-negative BC patients, indications for SLNB were extended to include patients with clinically node-negative (through palpation) breast cancer and up to two histopathologically confirmed macrometastatic SLNs as a surgical staging procedure, according to the landmark ACOSOG Z0011 study. Patients were randomized to undergo ALND or no further surgery, showing no differences in locoregional recurrence, disease-free survival (DFS), or overall survival (OS) on long-term follow-up [18]. Several validation studies have confirmed these results, addressed limitations of the Z0011 study, and the fostered omission of ALND in these patients [19,20,21,22,23,24] (Level of Evidence [LoE] I according to the Oxford Levels of Evidence 2 [25]). Most recently, results on the secondary endpoint recurrence-free survival from the randomized-controlled SENOMAC trial were published [26]. This study included 2766 patients with cT1-3 tumors, no palpably suspicious LNs, and one to two macrometastatic SLNs. Patients were eligible if they underwent BCS (63.8%) or mastectomy (36.2%), followed by adjuvant radiotherapy. Patients were randomized to undergo SLNB (*n* = 1335) or completion ALND (*n* = 1205). About one-third of patients had extranodal extension, and 89% received regional nodal irradiation (RNI). Recurrence-free survival showed no difference after a median follow-up of 37.1 months (hazard ratio 0.89, 95% confidence interval 0.65–1.20). Therefore, the omission of ALND is currently extended beyond the “Z0011 population” to include patients undergoing mastectomy, confirming previous results of a sub-analysis of the SINODAR-ONE trial [27], and those exhibiting extranodal disease when axillary radiotherapy is performed (LoE II).

These results prompted clinicians to ask three questions: (i) whether any surgical staging of the axilla is at all necessary; (ii) if completion ALND is necessary in patients with three or more positive SLNs; and (iii) if completion ALND is necessary in patients with clinically node-positive breast cancer.

### 2.1. Patients with Clinically and Imaging Node-Negative Breast Cancer

Recently, results from the randomized-controlled SOUND trial have been reported [33]. In this study, 1405 patients with clinically and imaging node-negative stage I BC were randomized to undergo SLNB (*n* = 708) or no axillary surgery (*n* = 697). The 5-year distant DFS was similar between both groups, showing a non-inferiority of non-surgical staging. Importantly, adjuvant treatment was similar between both groups; however, rates of axillary radiotherapy were not reported. Even though eligibility criteria encompassed patients with BC of all receptor subtypes, the main study population were postmenopausal patients with ER+/Her2− BC (87.8%). Therefore, the authors conclude that their results are predominantly applicable to this patient cohort (LoE I).

In light of the first question, previous results of a complete omission of any axillary surgery [3,4] as mentioned above are proving the concept, but both systemic and radiotherapeutic treatment regimens have undergone major developments [28,29]. However, the results of the CALGB 9343 trial suggest that in women ≥70 years of age with stage I estrogen-receptor (ER) positive, human epidermal growth factor receptor 2 (HER2) negative BC, the omission of surgical axillary staging does not result in worse oncologic outcomes [30]. Therefore, Choosing Wisely recommendations have advised against the routine use of SLNB in this cohort of patients, which is supported by current St. Gallen consensus recommendations [31,32]. Multiple trials have been designed to investigate whether patients with unremarkable palpatory examination of the axilla and no suspicious findings on axillary ultrasound can be spared any surgical staging [33,34,35,36,37].

Whilst results from the remaining trials on the omission of SLNB are pending, an important quest will be to integrate these results into real-world practice, as it was repeatedly shown that the Choosing Wisely recommendations were not followed in clinical practice [38,39,40]. Possible reasons for this are the categorization of SLNB as a low-risk procedure, the lost pathological staging information, and the partial lack of compatibility with inclusion criteria in some of the major trials on radiotherapy-omission and hypofractionation [41,42,43,44,45]. Therefore, interdisciplinary consensus recommendations seem paramount to integrate surgical de-escalation without unjustified adjuvant treatment escalation due to limited staging information in individual treatment plans [32].

### 2.2. Patients with Clinically Node-Negative Breast Cancer with More Than Two Positive Sentinel Lymph Nodes

The second question concerns clinically node-negative patients who do undergo SLNB and are shown to have three or more positive LNs. Whilst the Z0011 study showed a global impact with declining completion ALND (cALND) rates in patients meeting eligibility criteria [46,47,48,49,50,51,52,53,54], cALND is still recommended for those with three or more positive LNs [55,56].

In clinical practice however, cALND is increasingly omitted in about one-third to one-half of patients with three or more positive SLNs, who otherwise meet Z0011 eligibility criteria [57]. Associated factors with cALND omission were shown to be patient (older age), tumor (lower tumor grade), and treatment characteristics (BCS, no radiotherapy, the number of SLNs examined, the number of positive SLNs, and non-academic setting). Oncologic outcomes have only sparsely been reported for this cohort and are mainly stemming from retrospective cohort studies. Those, however, did not find differences in survival [58,59] (LoE III). Therefore, the omission of cALND in patients with three or more positive SLNs otherwise meeting Z0011 eligibility criteria cannot currently be recommended as the standard of care but may be considered for individual cases.

### 2.3. Patients with Clinically Node-Positive Breast Cancer

The third question is still quite original and under-addressed, at least in the adjuvant setting. Clinical nodal positivity is primarily defined by means of palpable nodal disease and was recently expanded to include non-palpable imaging-positive nodal disease. At least for the former, ALND is recommended outside of clinical studies. However, whilst both palpatory and ultrasound findings are indicative of a higher nodal tumor burden, almost half of these patients still show two or fewer involved nodes and would therefore qualify for ALND omission when applying the Z0011 criteria to this patient population with clinically node-positive BC [60,61,62,63]. Axillary ultrasound is examiner-dependent, with positive-predictive values between 58 and 81% and negative predictive values between 71 and 79%, yet it may aid in refining LN positivity prediction [60,61,62,64,65,66,67]. However, pathological nodal stage was not found to be higher in patients with palpable nodal disease compared to those with imaging-positive disease [63].

The OPBC-03/TAXIS trial is an ongoing, international, phase-III trial, investigating the non-inferiority of ART vs. ALND with respect to DFS for patients with clinically node-positive BC [68]. It is currently the only ongoing study investigating the de-escalation of axillary surgery in patients with clinically node-positive BC in the upfront surgery setting, but also includes patients with residual disease after NACT. As these are mainly patients with HR+/Her2− BC, for whom NACT currently shows pathological complete response (pCR) rates below 25% in clinical trials [69,70] and <20% in real-world settings [71], these patients represent a population with a significant unmet medical need. The applied surgical technique in the TAXIS study is tailored axillary surgery (TAS), which consists of the removal of the SLNs, all palpably suspicious LNs, and the clipped and pathologically confirmed metastatic LN, which can optionally be targeted under imaging-guidance [72]. The aim is to perform both a diagnostic staging procedure and a therapeutic removal of nodal disease. The idea is to reduce nodal tumor burden selectively to the point where radiotherapy can control it.

A pre-specified subproject of the TAXIS trial after the randomization of the first 500 patients showed that in the upfront surgery setting (*n* = 335), 88.4% (*n* = 296) had HR+/Her2− BC. Among those patients, a median of five LNs were resected during TAS, three of which were found to be metastatic, compared to nineteen, of which four were metastatic, during ALND. The main results showed that adjuvant systemic therapy decisions did not differ between patients with or without ALND [73] (LoE III). This was especially reassuring as adjuvant chemotherapy decisions traditionally still depend on the number of positive LNs in patients with luminal breast cancer. Chemotherapy is still indicated in patients with four or more positive LNs, with those patients being ineligible for trial protocols investigating genomic assays for chemotherapy decisions [74,75,76].

Genomic risk scores are also being used to assess whether omitting RNI is safe in patients with clinically node-positive BC or T3N0 BC who are ER+ and Her2− and have a recurrence score ≤ 25 as assessed by Oncotype Dx in the currently recruiting Tailor RT trial (NCT03488693). Therefore, biomarker-informed adjuvant radiotherapy decisions are beginning to focus on RNI following the publication of promising results and the initiation of several trials on the omission of breast radiotherapy in low-risk BC [45,77,78,79,80].

Adjuvant therapy decisions may also concern the recommendation for cyclin-dependent kinase 4/6 (CDK4/6) inhibitors for patients with higher-risk HR+/Her2− BC [81,82,83]. Whilst eligibility for Ribociclib is rather straight forward, including stage II and III BC, eligibility for the monarchE study, investigating the addition of adjuvant abemaciclib, includes patients with four or more positive LNs, or patients with one to three positive LNs and additional risk features (tumor ≥ 5 cm, histologic grade 3, Ki67 ≥ 20%). Therefore, the question arose, whether in patients with one to three positive LNs without additional risk factors, cALND should be performed to determine eligibility for adjuvant abemaciclib based on the cut-off of four or more positive LNs. Two retrospective cohort studies, including 2299 patients, found that cALND would constitute a surgical overtreatment for 87% (*n* = 1999) of those patients, who were not found to have four or more positive LNs [84,85] (LoE III). Therefore, cALND is currently not recommended as a standard of care to inform adjuvant abemaciclib treatment [32,86].

## 3. Axillary Surgery in the Neoadjuvant Setting

NACT poses the opportunity for an in vivo drug-sensitivity testing and has seen major advances over the past decades [69,70,87,88,89,90,91]. Surgically, rates of BCS can be improved, and current studies investigate the role of vacuum-assisted biopsy as an alternative to surgical excision in exceptional clinical responders [90,92,93,94]. In the post-neoadjuvant setting, axillary surgery still only provides two treatment options: the confirmation of pCR through SLNB or targeted lymph node removal, or cALND in the case of residual disease. However, these paradigms are currently being challenged. Importantly, NACT was found to be an independent predictor of lymphedema after ALND [11]. Therefore, investigations on oncologic outcomes with tailored surgical approaches after NACT to reduce surgery-related morbidity address an important medical need.

### 3.1. Clinically Node-Negative Patients

Indications for NACT were extended from initially locally advanced BC to cases with node-negative disease [87,88]. In the case of clinically node-negative status pre-NACT and sustained clinical node negativity after NACT (ycN0), SLNB is considered oncologically sufficient to confirm pathological node negativity (ypN0) [95,96,97,98]. However, if SLNs are shown to be metastatic, cALND is generally recommended [32,55,56].

Nodal metastases are very infrequently encountered in these patients, and pCR rates of the breast correlate well with axillary pCR rates [99,100,101]. A retrospective cohort analysis showed that among cN0 patients with breast pCR, the rate of axillary LN metastasis as assessed by ALND was 0% [102] (LoE III). In a Dutch retrospective cohort study, almost all patients with triple-negative BC (TNBC) and Her2 positive subtype and a radiological complete response (rCR) had no positive axillary LNs as assessed through SLNB [103] (LoE III). A retrospective multicenter study from the UK showed an association of rCR as assessed through mammography, ultrasound, and MRI with ypN0 status irrespective of molecular subtype [104] (LoE III). This lay the ground for two currently ongoing clinical trials, prospectively investigating the omission of axillary surgery in clinically node-negative patients with TNBC or Her2 positive BC and rCR after NACT [105,106].

### 3.2. Patients with Clinically Node-Positive Breast Cancer Who Are Rendered Node-Negative after NACT

Neoadjuvant chemotherapy has the potential to locoregionally eradicate disease. Nodal pCR rates in patients with initially node-positive BC range from <20% in HR+/Her2− up to approximately 60–70% in Her2+ and TNBC patients [71,107]. Therefore, quests to de-escalate axillary surgery were undertaken. Primarily, efforts concentrated on identifying the false-negative rate (FNR) of SLNB in these patients, which was found to decrease when following certain rules (Table 1): when three or more SLNs were retrieved and/or dual tracer mapping was used, FNR was found below the commonly accepted, yet arbitrary, threshold of 10%. In a next step, surgical approaches with targeted lymph node removal were applied, including (i) targeted axillary dissection (TAD), incorporating the imaging-guided retrieval of one clipped and histologically-confirmed positive LN and all SLNs, which shows a FNR of 2–12% [108]; (ii) the MARI procedure, in which the largest tumor-positive LN is marked with a radioactive seed before NACT and retrieved thereafter, which shows a FNR of 7%; and (iii) the RISAS approach, combining SLNB and the MARI procedure, with a FNR of 3.5% [109,110,111] (Table 1). Oncologically, the first three approaches showed comparable outcomes, with regional recurrence rates below 3% over follow-up periods of up to 9 years [96,107,112,113,114,115,116,117,118,119]. However, it was unclear whether any of the two most commonly performed procedures—SLNB or TAD—showed any oncologic benefits. To address this question, the international multicenter retrospective OPBC-04/OMA study included 1144 patients, who underwent either SLNB (*n* = 666) or TAD (*n* = 478). Whilst more LNs were retrieved using SLNB (median *n* = 4) compared to TAD (median *n* = 3), both techniques showed very low axillary recurrence rates (3-year incidence of axillary recurrence: SLNB 0.5%, TAD 0.8%, *p* = 0.55). No difference in other oncologic outcomes was seen, supporting the omission of ALND with either technique. It will be of the utmost interest to see whether the two ongoing prospective randomized trials investigating the omission of ALND and/or nodal irradiation in patients with complete nodal response will confirm these results [120,121]. In the NRG oncology/NSABP B-51/RTOG 1304 trial, which was presented at San Antonio Breast Cancer Symposium (SABCS) 2023, patients who were found node-negative through SLNB and/or ALND after NACT were randomized 1:1 to undergo RNI or not. Among 1556 evaluable patients, no differences in 5-year invasive DFS were found (91.8% without RNI vs. 92.7% with RNI) [122].

### 3.3. Patients with Clinically Node-Positive Breast Cancer and Residual Nodal Disease

The recommended surgical procedure for patients with residual nodal disease after NACT remains ALND [32,55,56]. However, the omission of ALND is becoming more frequent and increasingly accepted amongst clinicians despite limited evidence on oncologic outcomes [32,131]. The probability of additional positive nodes seems to be related to the type of residual nodal disease, which may enable a more differentiated approach.

#### 3.3.1. Isolated Tumor Cells

Until recently, only limited information on patients with isolated tumor cells (ITCs) existed in the literature. The international multicenter retrospective OPBC-05/ICARO study was presented at SABCS 2023 and investigated 583 patients with residual ITCs [132]. Of those, 182 underwent ALND and 401 did not. Additional positive nodes were found in 30% of patients undergoing cALND. No difference in the 5-year rate of any axillary recurrence (ALND: 1.7%, no ALND: 1.1%, *p* = 0.7) or in any other oncologic endpoint was found. Therefore, the routine use of ALND in this population does not seem to be warranted.

#### 3.3.2. Patients with Residual Nodal Micro- and Macrometastases

In micro- and macrometastatic residues, additional positive nodes are found in 60% of patients, and the likelihood of additional positive nodes is not associated with BC receptor subtype [133,134,135]. Three randomized controlled trials, the Alliance A011202 trial, the ADARNAT trial, and the OPBC-03/TAXIS trial are currently investigating whether ART is non-inferior to ALND in this population [68,136,137].

The Alliance A011202 trial includes patients with confirmed LN metastasis at diagnosis (cN1), who are clinically rendered node-negative (ycN0), yet have residual micro- or macrometastatic nodal disease (ypN+) upon SLNB after NACT. Patients are randomized to undergo ART or ALND. Enrollment is finalized with 2010 patients included in the study. The results are awaited around 2030.

The Spanish ADARNAT trial includes patients with clinically node-negative or node-positive BC with nodal disease in one to two sentinel nodes after NACT or neoadjuvant endocrine treatment. These patients are randomized to ART or ALND, with the primary endpoint being the non-inferiority of axillary recurrence. A pilot phase including 100 patients was finalized in April 2023, and the accrual of the planned total of 1660 patients is ongoing [138].

After NACT, residual nodal disease has to be confirmed for patients to be eligible for the TAXIS trial. Dual-tracer mapping is recommended, whilst imaging guidance to selectively remove the clipped and histopathologically confirmed positive LN is optional. However, specimen radiography is included as an obligatory quality assurance measure with the mandatory excision of the clipped node. The results from a prespecified subproject of the first 500 randomized patients within the TAXIS study show that a larger proportion of patients with Her2+ BC or TNBC underwent NACT and that NACT administration increased over the study period [139] (LoE III). Importantly, as in the upfront surgery setting, also after NACT, neither the proportion of patients undergoing adjuvant therapy nor the type of post-neoadjuvant treatment differed between patients who underwent TAS vs. ALND [73] (LoE III). It was reassuring to confirm that the FNR of TAS—being 2.6%—did not lead to differences in response-driven post-neoadjuvant therapy [63]. The accrual of the OPBC-03/TAXIS trial currently shows a completion of two-thirds of the total planned sample size of 1500 patients (Figure 1). Accrual completion is projected for the end of 2025. The primary endpoint analysis is expected in 2030.

In the meantime, retrospective real-world studies using routinely collected data and substudies from completed trials are being performed. Multiple, rather small institutional series have reported on patients with residual micro- or macrometastases and found axillary nodal recurrence rates between 0 and 28.6% over follow-up periods extending up to 9.2 years [96,107,113,114,116,117,119,140] (LoE III). Data from the NSABP B-40 and NSABP B-41 trial were used to infer oncologic outcomes amongst patients with residual nodal disease, who underwent either SLNB, SLNB and cALND, or ALND. Amongst 630 ypN+ patients, of which 51% received RNI, SLNB was not associated with locoregional recurrence, distant recurrence, disease-free survival, or overall survival compared to cALND and ALND [141] (LoE III). The MARI study also includes patients with residual disease in the MARI node, who either undergo ART if fewer than four nodes are found to be FDG-avid on staging FDG PET-CT pre-NACT, or cALND in the case of four or more suspicious nodes. While the 3-year recurrence-free survival was excellent at 98.2%, all five recorded recurrences occurred in patients with TNBC who did not undergo cALND [107] (LoE III). Some authors therefore argue that ART may be suboptimal for patients with residual nodal disease and triple-negative subtype [142]. The OPBC-07/microNAC study is currently in preparation and will include patients with residual nodal micrometastases after NACT in an international multicenter retrospective cohort. The primary endpoint will be the 5-year rate of any axillary recurrence using ALND.

In 2024, ALND remains the standard of care for patients with residual nodal disease after NACT. The omission of ALND may be safe in patients with ITCs as assessed in a global retrospective cohort study, whilst foregoing ALND in patients with micro- or macrometastatic disease should currently only be advised within clinical trials. Caution should be taken to change clinical practice following results from informative clinical studies.

## 4. Axillary Surgery in Special Situations

### 4.1. Inflammatory Breast Cancer

Inflammatory BC (IBC) represents a rare but aggressive subset of BC. The 5-year overall survival rate in non-metastatic IBC is 59%, with locoregional recurrences being reportedly similar to non-IBC if contemporary trimodal therapy, consisting of neoadjuvant chemotherapy, total mastectomy including ALND, and adjuvant radiotherapy, is followed [32,55,56,143,144]. However, only a minority of patients does follow the recommended therapy according to a study from the US National Cancer Database (NCDB), in which pathological nodal status was found to be prognostic [145]. Another study using the NCDB found that the removal of ≥10 LNs showed improved OS in cN2-3 patients, but not in cN0, suggesting that in the latter, ALND may represent an overtreatment with associated surgical morbidity [146] (LoE III). Nevertheless, in clinical practice, ALND remains the standard of care in IBC [32,55,56].

### 4.2. Locoregional Recurrence

In cases of suspected locoregional recurrence (LRR), the exclusion of a new ipsilateral tumor, staging, and the determination of receptor status should be performed to guide treatment [55,147]. Evidence on ideal axillary surgery in locoregional recurrence is scarce. In a retrospective multicenter Dutch cohort study of patients with axillary recurrence performed between 2002 and 2004, Bulte et al. found that 5-year post-recurrence overall survival was 58%, with axillary treatment not having a significant impact on survival [148] (LoE III). A retrospective population-based Canadien series of patients with axillary recurrence performed between 1989 and 2003 showed that 5-year post-recurrence overall survival was 49.3%. In this study, 26.8% received no axillary surgery, 47.3% underwent isolated lymph node removal including SLNB, and 25.9% underwent ALND. The extent of performed axillary surgery (isolated lymph node removal vs. ALND) did not impact survival [149] (LoE III). In clinical practice, the omission of axillary surgery in clinically node-negative LRR is advocated for by some, arguing that systemic treatment decisions are mainly based on tumor biology [150,151]. The prospective randomized CALOR trial established the role of chemotherapy in patients with LRR, showing no benefit in regard to DFS in patients with ER-positive disease, while being beneficial for those with ER-negative LRR [151]. The POLAR trial is an ongoing study investigating the role of adjuvant palbociclib in HR+/HER2− LRR [152]. Re-SLNB is technically feasible, and supporters argue that adjuvant radiotherapy may be impacted by it [153,154,155]. Whether ALND needs to be performed in cases of unsuccessful repeat SLNB was investigated in a Dutch national registry. In patients who underwent ALND, 13% had metastatic lymph nodes; however, 5-year regional recurrence rates did not differ in patients with vs. without ALND [156] (LoE III). ALND is indicated in patients with clinically node-positive recurrence.

## 5. Conclusions

Efforts to de-escalate axillary surgery in BC have led to major changes in treatment paradigms, sparing thousands of women the associated morbidity of ALND whilst remaining oncologically safe, as shown in randomized controlled trials with a high level of evidence. Few indications for ALND remain, with real-world studies being indicative of safe omission possibilities even if ITCs are found after NACT. Such studies generally have a lower level of evidence. However, they can address clinically relevant questions in a global manner, quickly, with the largest number of participants, and therefore the strongest statistical power available. For patients with clinically node-positive BC, the results of three randomized controlled trials—namely, the Alliance A011202 trial, the ADARNAT trial, and the OPBC-03/TAXIS trial—are eagerly awaited.

## Figures and Tables

**Figure 1 cancers-16-01623-f001:**
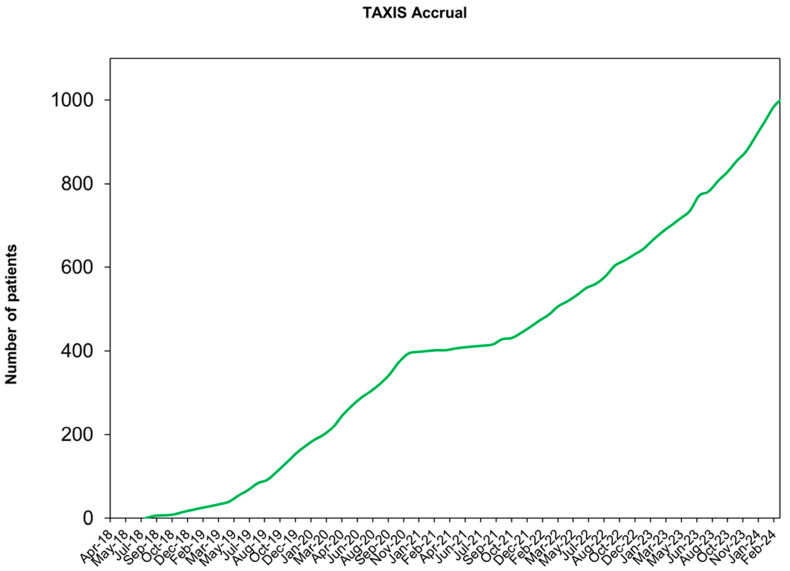
Accrual of the TAXIS study.

**Table 1 cancers-16-01623-t001:** False-negative rates of sentinel lymph node biopsy and targeted lymph node removal compared to completion axillary lymph node dissection in patients with initially clinically node-positive breast cancer rendered clinically node-negative following neoadjuvant chemotherapy.

First Author Study/Site (Year of Publication)	Study Design and Setting	Study Period	*n*	Inclusion Criteria	Sampled-Node Marked	Axillary Surgery	Axillary Tracer	Number of Removed Nodes	Axillary Dissection	Lymph Node Positivity Including Isolated Tumor Cells	False-Negative Rate	False-Negative Rate after Removal of Three or More LNs
Sentinel lymph node biopsy												
Shen (2007) MD Anderson [123]	Retrospective Single-center Tertiary	1994–2002	56	cT1-4, cN1-3 NACT	No	SLNB	83% dual tracer	Median: 2	100%	nR	25%	nR
Alvarado (2012) MD Anderson [124]	Retrospective Single-center Tertiary	1994–2010	111	cT1-4, cN1-3 NACT	No	SLNB	77% dual tracer	Median: 2 Mean: 2.6	100%	nR	20.8%	nR
Boughey (2013) ACOSOG Z1071 [125]	Prospective Multicenter National	07/2009–06/2011	756	cT0-4, cN1-2 NACT	Not mandatory	SLNB	79% dual tracer 17% Tc 4% blue dye	Median: 2	100%	No	12.6%	9.1%
Kuehn (2013) SENTINA [126]	Prospective Multicenter Germany, Austria	09/2009–05/2012	592	cN+ NACT	Not mandatory	SLNB	67% single tracer	Median: 2 Mean: 2.7	100%	nR	14.2%	≤7.3%
Boileau (2015) SN-FNAC [127]	Prospective Multicenter USA, Canada	03/2009–12/2012	153	cT0-3; cN1-2 NACT	nR	SLNB	Dual tracer recommended; Tc mandatory	Mean: 2.7	100%	Yes	13.3% (incl. ypN0(i+)) 8.4% (excl. ypN0(i+))	≤4.9%
Caudle (2016) MD Anderson [108]	Prospective Single-center Tertiary	2011–2015	191	cN+ NACT	Yes	SLNB	55% dual tracer	Mean: 2.7	100%	Yes	10.1%	10.3%
Martelli (2017) IRCCS Milan [128]	Retrospective Single-center Tertiary	01/2002–12/2007	139	cT2, cN0-1 NACT	No	SLNB	100% single tracer (Tc)	Median: 2	100%	nR	11.3%	0%
Classe (2019) GANEA-2 [98]	Prospective Multicenter National	07/2010–07/2014	307	cT1-3, cN1-2 NACT	No	SLNB	Dual tracer recommended	Median: 2	100%	No	11.9%	≤7.8%
Targeted lymph node removal												
Donker (2015) Netherlands Cancer Institute [109] MARI	Retrospective Single-center Tertiary	10/2008–11/2012	100	cN+ NACT	Yes	MARI	Iodine Seed	1	100%	Yes	7%	nA
Caudle (2016) MD Anderson [108]	Prospective Single-center Tertiary	2011–2015	191	cN+ NACT	Yes	TAD	Iodine Seed	nR	100%	Yes	Marked node only: 4.2% TAD 2.0%	nR
Simons (2022) RISAS [129]	Prospective Multicenter Netherlands, UAE	03/2017–12/2019	212	cT1-4, cN1,2,3b NACT	Yes	RISAS (SLNB + MARI node)	Iodine Seed	Mean: 1.8 Median: 2	100%	Yes	SLNB only: 17.9% Marked node only: 7.0% RISAS: 3.5%	nR
Kuemmel (2023) SENTA [130]	Prospective Multicenter National	01/2017–10/2018	199	cT1-4, cN+ NACT	Yes	TAD	SLNB using single or dual tracer (Tc, dye) Imaging Localization	Median: 3	40.2% (80/199)	Yes	4.2%	nR
Wu (2023) Fudan University Shanghai [119]	Retrospective Single-center Tertiary	03/2014–04/2021	152	cT1-4, cN1-3 18–70 years	Yes	TAD	72% Single tracer Localization	nR	100%	Yes	12.2%	13.0%

NACT—neoadjuvant chemotherapy; Tc—Technetium; SLNB—sentinel procedure; TAD—targeted axillary dissection; MARI—marking axillary lymph nodes with radioactive iodine seeds; RISAS—radioactive iodine seed placement in the axilla with sentinel lymph node biopsy; nR—not reported; nA—not available.

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
