# Peer review of "Axillary Surgery for Breast Cancer in 2024"

_cancers, 2024, doi:10.3390/cancers16091623_

Round 1

Reviewer 1 Report

Comments and Suggestions for Authors

This narrative review about axillary surgery for breast cancer in 2024 performed by Heidinger M and Weber WP depicts the evolution of axillary surgery (de-evolution?), highlighting the shift from radical dissection to a more individualized approach.

The manuscript shows evidence supporting de-escalation of axillary surgery, citing major studies such as NSABP-04, ACOSOG Z0011, and the SENOMAC trial.

These studies demonstrated that less invasive procedures like sentinel lymph node biopsy (SLNB) do not compromise oncological outcomes while significantly reducing morbidity.

I just have some minor comments:

- Introduction and Rationale, sections 1 and 2, can be condensed together;

- In section 3, lines 70-81, the authors presented the latest results of the SENOMAC trial which was presented at the latest SABCS, which showed that omission of ALND is currently extended beyond the “Z0011 population” to include patients undergoing mastectomy. In this regard, I suggest to cite the study PMID: 37471574 which actually presents the results of a sub-analysis of the SINODAR-ONE trial, only showing the outcomes of patients treated with mastectomy with 1-2 positive sentinel lymph nodes;

- The last paragraph, 5.2, is too brief. Either you expand it or remove it.

Author Response

This narrative review about axillary surgery for breast cancer in 2024 performed by Heidinger M and Weber WP depicts the evolution of axillary surgery (de-evolution?), highlighting the shift from radical dissection to a more individualized approach.

The manuscript shows evidence supporting de-escalation of axillary surgery, citing major studies such as NSABP-04, ACOSOG Z0011, and the SENOMAC trial.

These studies demonstrated that less invasive procedures like sentinel lymph node biopsy (SLNB) do not compromise oncological outcomes while significantly reducing morbidity.

I just have some minor comments:

- Introduction and Rationale, sections 1 and 2, can be condensed together;

We thank the reviewer for this suggestion, which we gladly adopted condensing sections 1 and 2 as follows:

Axillary surgery for breast cancer (BC) has evolved significantly from a previous “one size fits all” approach that involved radical surgery, including lymph node dissection extending from the axilla to the neck, to an increasingly granular and individualized surgical treatment. Axillary lymph node dissection (ALND) was the standard of care for all patients with BC until the nineties, which was considered to be a therapeutic procedure. The rationale behind it was that complete surgical removal of locoregional tumor residues would result in improved survival. A hypothesis that has never been proven and was already questioned by the NSABP-04 trial. In this landmark study, patients with clinically node-negative and node-positive BC were shown to have similar 10-year overall survival outcomes no matter whether ALND or axillary radiotherapy (ART) were performed. These results could be confirmed in clinically node-negative patients, who underwent breast-conserving surgery (BCS) and adjuvant radiotherapy of the breast. Therefore, radical surgery and its associated morbidity was increasingly questioned. Axillary staging information was still deemed necessary, leading to the development of the sentinel lymph node (SLN) biopsy (SLNB). Whilst showing a false-negative rate of around 10%, excellent oncological outcomes were achieved. Notably, SLNB dramatically reduced surgical morbidity and improved quality of life. Nevertheless, approximately 5% of patients still experience surgery-related morbidity. Therefore, studies to identify patients, in whom surgical axillary staging can be abandoned altogether have been initiated.

In the present manuscript, current evidence on axillary surgery for BC in the upfront surgical setting, after neoadjuvant chemotherapy (NACT) and in special situations such as inflammatory BC and locoregional recurrence are reviewed.

- In section 3, lines 70-81, the authors presented the latest results of the SENOMAC trial which was presented at the latest SABCS, which showed that omission of ALND is currently extended beyond the “Z0011 population” to include patients undergoing mastectomy. In this regard, I suggest to cite the study PMID: 37471574 which actually presents the results of a sub-analysis of the SINODAR-ONE trial, only showing the outcomes of patients treated with mastectomy with 1-2 positive sentinel lymph nodes;

We thank the reviewer for this suggestion. We have now included the reference of the SINODAR-ONE trial in the respective paragraph, which reads as follows:

Therefore, omission of ALND is currently extended beyond the “Z0011 population” to include patients undergoing mastectomy, confirming previous results of a sub-analysis of the SINODAR-ONE trial, and those exhibiting extranodal disease when axillary radiotherapy is performed (Level of Evidence [LoE] II).

- The last paragraph, 5.2, is too brief. Either you expand it or remove it.

We agree with the reviewer that the section on locoregional recurrence can be expanded. The paragraph now reads as follows:

In cases of suspected locoregional recurrence (LRR) exclusion of a new ipsilateral tumor, staging and determination of receptor status should be performed to guide treatment. Evidence on ideal axillary surgery in locoregional recurrence is scarce. In a retrospective multicenter Dutch cohort study of patients with axillary recurrence performed between 2002 and 2004, Bulte et al. found that 5-year post-recurrence overall survival was 58% with axillary treatment not having a significant impact on survival (LoE III). A retrospective population-based Canadien series of patients with axillary recurrence performed between 1989 and 2003 showed that 5-year post-recurrence overall survival was 49.3%. In this study 26.8% received no axillary surgery, 47.3% isolated lymph node removal including SLNB, and 25.9% underwent ALND. The extent of performed axillary surgery (isolated lymph node removal vs. ALND) did not impact survival (LoE III). In clinical practice, omission of axillary surgery in clinically node-negative LRR is advocated for by some, arguing that systemic treatment decisions are mainly based on tumor biology. The prospective randomized CALOR trial established the role of chemotherapy in patients with LRR, showing no benefit in regard to DFS in patients with ER-positive disease, while being beneficial for those with ER-negative LRR. The POLAR trial is an ongoing study investigating the role of adjuvant palbociclib in HR+/HER2- LRR. Re-SLNB is technically feasible, and supporters argue that adjuvant radiotherapy may be impacted by it. Whether ALND needs to be performed in cases of unsuccessful repeat SLNB was investigated in a Dutch national registry. In patients who underwent ALND 13% had metastatic lymph nodes, however 5-year regional recurrence rates did not differ in patients with vs. without ALND (LoE III). ALND is indicated in patients with clinically node-positive recurrence.

Reviewer 2 Report

Comments and Suggestions for Authors

The review covers completely the current important topic in axillary surgery in breast cancer. It is well structured and clearly written. The scientific content  is very well explained und discussed. The cited references are appropiate and contain mostly recent publications. An excessive number of self-citations is not found. In summary, no corrections are necessary. 

Author Response

We thank the reviewer for the review of the manuscript and its contextualization.

Reviewer 3 Report

Comments and Suggestions for Authors

- The review is pretty comprehensive focussing on the value of de-escalating axillary surgery in patients with primary breast cancer in the setting of both upfront surgery and neoadjuvant chemotherapy.  It does not add to much to existing literature.  To make it worthy of publication in this special issue of Cancers, I suggest the following revisions.

(i) Assign the level of evidence based on the hierarchy of evidence for clinical trials to key studies (or at least the study with the highest level of evidence to date) under each category of discussion.

(ii) Elaborate the role of (with evidence as above) axillary radiotherapy in contexts where axillary surgery may be de-escalated.

(iii) What are the specific contexts, based on evidence (see above), when de-escalation is appropriate?  Can it be applicable to 1) all situations; 2) just for certain tumour biology (e.g. luminal A); 3) under certain context (e.g. receiving endocrine therapy or a systemic therapy); 4) certain patient population (e.g. frail, older patients with limited life expectancy)?

(iv) Make distinction when discussing the studies presented between RCTs with high level of evidence and real-life data (the authors mention quite a lot about this category).

(v) The conclusion should be amended to reflect the above, in order not to mislead practising clinicians so that they may simply change their clinical practice without understanding the level of evidence currently available.  It would be helpful then to mention which trials are pivotal to await results of, in order to definitely change current practice as appropriate.  

- Minor suggestion re: language - Please avoid saying 'node positive patients' but say 'patients with node positive cancer' instead.  The comment applies to other similar situations throughout. 

Author Response

The review is pretty comprehensive focussing on the value of de-escalating axillary surgery in patients with primary breast cancer in the setting of both upfront surgery and neoadjuvant chemotherapy.  It does not add to much to existing literature.  To make it worthy of publication in this special issue of Cancers, I suggest the following revisions.

(i) Assign the level of evidence based on the hierarchy of evidence for clinical trials to key studies (or at least the study with the highest level of evidence to date) under each category of discussion.

We thank the reviewer for directing us to this important point. As suggested, we have included the level of evidence according to the Oxford Centre for Evidence-Based Medicine 2011 Levels of Evidence (OCEBM Levels of Evidence Working Group. The Oxford Levels of Evidence 2 Available from: https://www.cebm.ox.ac.uk/resources/levels-of-evidence/ocebm-levels-of-evidence) for each peer-reviewed and at the time of writing published key study on axillary surgery under each category of discussion. Therefore, we have made the following changes in the manuscript:

Category - Axillary surgery in the upfront surgery setting

Several validation studies have confirmed these results, addressed limitations of the Z0011 study, and fostered omission of ALND in these patients (Level of Evidence [LoE] I).

Therefore, omission of ALND is currently extended beyond the “Z0011 population” to include patients undergoing mastectomy, confirming previous results of a sub-analysis of the SINODAR-ONE trial, and those exhibiting extranodal disease when axillary radiotherapy is performed (LoE II).

Category - Patients with clinically and imaging node-negative breast cancer

Even though eligibility criteria encompassed patients with BC of all receptor subtypes, the main study population were postmenopausal patients with ER+/Her2- BC (87.8%). Therefore, the authors conclude that their results are predominantly applicable to this patient cohort (LoE I).

Category - 2.2. Patients with clinically node-negative breast cancer with >2 positive sentinel lymph nodes

Oncological outcomes have only sparsely been reported for this cohort and are mainly stemming from retrospective cohort studies. Those did however not find differences in survival (LoE III).

Category - 2.3. Patients with clinically node-positive breast cancer

The main results showed that adjuvant systemic therapy decisions did not differ between patients with or without ALND (LoE III).

Two retrospective cohort studies, including 2299 patients found that cALND would constitute a surgical overtreatment for 87% (n=1999) of those patients, who were not found to have ≥4 positive LNs 84,85 (LoE III).

Category – 3.1. Clinically node-negative patients

A retrospective cohort analysis showed that among cN0 patients with breast pCR the rate of axillary LN metastasis as assessed by ALND was 0% (LoE III). In a Dutch retrospective cohort study, almost all patients with triple-negative BC (TNBC) and Her2 positive subtype and a radiological complete response (rCR) had no positive axillary LNs as assessed by SLNB (LoE III). A retrospective multicenter study from the UK showed an association of rCR as assessed by mammography, ultrasound, and MRI with ypN0 status irrespective of molecular subtype (LoE III).

Category – 3.3.2 Patients with residual nodal micro- and macrometastases

Results from a prespecified subproject of the first 500 randomized patients within the TAXIS study show that a larger proportion of patients with Her2+ BC or TNBC underwent NACT and that NACT administration increased over the study period (LoE III). Importantly, as in the upfront surgery setting also after NACT neither the proportion of patients undergoing adjuvant therapy nor the type of post-neoadjuvant treatment differed between patients that underwent TAS vs. ALND (LoE III).

Multiple, rather small institutional series have reported on patients with residual micro- or macrometastases and found axillary nodal recurrence rates between 0-28.6% over follow-up periods extending up to 9.2 years (LoE III).

Amongst 630 ypN+ patients, of which 51% received RNI, SLNB was not associated with locoregional recurrence, distant recurrence, disease-free survival, or overall survival compared to cALND and ALND (LoE III).

While the 3-year recurrence-free survival was excellent at 98.2%, all 5 recorded recurrences occurred in patients with TNBC who did not undergo cALND (LoE III).

Category - 4.1. Inflammatory breast cancer

Another study using the NCDB found that removal of ≥10 LNs showed improved OS in cN2-3 patients, but not in cN0, suggesting that in the latter ALND may represent an overtreatment with associated surgical morbidity (LoE III).

Category – 4.2 Locoregional recurrence

In a retrospective multicenter Dutch cohort study of patients with axillary recurrence performed between 2002 and 2004, Bulte et al. found that 5-year post-recurrence overall survival was 58% with axillary treatment not having a significant impact on survival (LoE III).

The extent of performed axillary surgery (isolated lymph node removal vs. ALND) did not impact survival (LoE III). 

In patients who underwent ALND 13% had metastatic lymph nodes, however 5-year regional recurrence rates did not differ in patients with vs. without ALND (LoE III).

(ii) Elaborate the role of (with evidence as above) axillary radiotherapy in contexts where axillary surgery may be de-escalated.

We thank the reviewer for this important suggestion. We have expanded the manuscript regarding the role of axillary radiotherapy in context of de-escalated axillary surgery as follows:

Category 2.3 – Patients with clinically node-positive breast cancer

Genomic risk scores are also being used to assess whether omitting RNI is safe in patients with clinically node-positive BC or T3N0 BC who are ER+ and Her2- and have a recurrence score ≤25 as assessed by Oncotype Dx in the currently recruiting Tailor RT trial (NCT03488693). Therefore, biomarker-informed adjuvant radiotherapy decisions are beginning to focus on RNI following the publication of promising results and the initiation of several trials on the omission of breast radiotherapy in low risk BC.

Category – 3.2. Patients with clinically node-positive breast cancer who are rendered node-negative after NACT

In the NRG oncology/NSABP B-51/RTOG 1304 trial, which was presented at SABCS 2023 patients who were found node-negative through SLNB and/or ALND after NACT were randomized 1:1 to undergo RNI or not. Among 1556 evaluable patients no differences in 5-year invasive DFS were found (91.8% without RNI vs. 92.7% with RNI).

(iii) What are the specific contexts, based on evidence (see above), when de-escalation is appropriate?  Can it be applicable to 1) all situations; 2) just for certain tumour biology (e.g. luminal A); 3) under certain context (e.g. receiving endocrine therapy or a systemic therapy); 4) certain patient population (e.g. frail, older patients with limited life expectancy)?

We agree with the reviewer that literature exists, defining certain subgroups of patients with breast cancer that seem to be more appropriate for de-escalation of axillary surgery than others. Therefore, we have e.g. included the exact eligibility criteria for the CALGB 9343 trial in the manuscript, which lay the ground for the Choosing wisely recommendations, as mentioned in the manuscript. Other trials, as e.g. the SENOMAC trial had other eligibility criteria as compared to the one specified by the reviewer, which is why only those are included in the manuscript. However, we agree with the reviewer that for certain studies, further information on subgroups mentioned by the reviewer is available. Where applicable, we have therefore included information on these subgroups in the manuscript, making the following adjustments:

          Category - 2.1. Patients with clinically and imaging node-negative breast cancer

Even though eligibility criteria encompassed patients with BC of all receptor subtypes, the main study population were postmenopausal patients with ER+/Her2- BC (87.8%). Therefore, the authors conclude that their results are predominantly applicable to this patient cohort (LoE I).

          Category – 2.3 Patients with clinically node-positive breast cancer

Genomic risk scores are also being used to assess whether omitting RNI is safe in patients with clinically node-positive BC or T3N0 BC who are ER+ and Her2- and have a recurrence score ≤25 as assessed by Oncotype Dx in the currently recruiting Tailor RT trial (NCT03488693). Therefore, biomarker-informed adjuvant radiotherapy decisions are beginning to focus on RNI following the publication of promising results and the initiation of several trials on the omission of breast radiotherapy in low risk BC.

Category - 3.3.2 Patients with residual nodal micro- and macrometastases

The MARI study also includes patients with residual disease in the MARI node, who either undergo ART if <4 nodes are found to be FDG-avid on staging FDG PET-CT pre-NACT, or cALND in case of ≥4 suspicious nodes. While the 3-year recurrence-free survival was excellent at 98.2%, all 5 recorded recurrences occurred in patients with TNBC who did not undergo cALND (LoE III). Some authors therefore argue that ART may be suboptimal for patients with residual nodal disease and triple-negative subtype.

(iv) Make distinction when discussing the studies presented between RCTs with high level of evidence and real-life data (the authors mention quite a lot about this category).

The reviewer is right that this distinction can be made clearer in the manuscript. As suggested in the first comment by the reviewer, we have included the level of evidence for each peer-reviewed and at the time of writing published key study on axillary surgery under each category of discussion, which should help the reader to incorporate the presented information better.

Furthermore, we have adapted the wording concerning the one presented real-world study to read as follows:

Category – 3.3.2 Patients with residual nodal micro- and macrometastases

In the meantime, retrospective real-world studies using routinely collected data and substudies from completed trials are being performed.

Lastly, we incorporated the reviewers’ comment in the conclusion, which now reads as follows:

Efforts to de-escalate axillary surgery in BC have led to major changes in treatment paradigms, sparing thousands of women the associated morbidity of ALND whilst remaining oncologically safe as shown in randomized controlled trials with a high level of evidence. Few indications for ALND remain, with real-world studies being indicative of safe omission possibilities even if ITCs are found after NACT. Such studies generally have a lower level of evidence. However, they can address clinically relevant questions in a global manner, quickly, with the largest number of participants and therefore the strongest statistical power available.

(v) The conclusion should be amended to reflect the above, in order not to mislead practising clinicians so that they may simply change their clinical practice without understanding the level of evidence currently available.  It would be helpful then to mention which trials are pivotal to await results of, in order to definitely change current practice as appropriate.  

We have amended the conclusion as partially outlined in the comment above to address this important comment by the reviewer. The complete conclusion now reads as follows:

Efforts to de-escalate axillary surgery in BC have led to major changes in treatment paradigms, sparing thousands of women the associated morbidity of ALND whilst remaining oncologically safe as shown in randomized controlled trials with a high level of evidence. Few indications for ALND remain, with real-world studies being indicative of safe omission possibilities even if ITCs are found after NACT. Such studies generally have a lower level of evidence. However, they can address clinically relevant questions in a global manner, quickly, with the largest number of participants and therefore the strongest statistical power available. For patients with clinically node-positive BC, the results of three randomized controlled trials – namely the Alliance A011202 trial, the ADARNAT trial, and the OPBC-03/TAXIS trial – are eagerly awaited.

- Minor suggestion re: language - Please avoid saying 'node positive patients' but say 'patients with node positive cancer' instead.  The comment applies to other similar situations throughout. 

We have adapted the wording as suggested by the reviewer throughout the manuscript.

Round 2

Reviewer 3 Report

Comments and Suggestions for Authors

The authors have adequately addressed my comments.  It would be helpful if they could make the following minor revisions in keeping with the changes made:

(1) Revise the abstract accordingly to reflect the impact of the level of evidence and subgroups.

(2) Insert a reference on the system of level of evidence used as appropriate.

Author Response

Reviewer 3

The authors have adequately addressed my comments.  It would be helpful if they could make the following minor revisions in keeping with the changes made:

(1) Revise the abstract accordingly to reflect the impact of the level of evidence and subgroups.

We agree with the reviewer that, for consistency, the abstract should be revised to reflect the changes in the main text. We have therefore adapted the abstract accordingly to read as follows:

Axillary surgery for patients with breast cancer (BC) in 2024 is becoming increasingly specific, moving away from the previous 'one size fits all' radical approach. The goal is to spare morbidity whilst maintaining oncologic safety. In the upfront surgery setting, a first landmark randomized controlled trial (RCT) on the omission of any surgical axillary staging in patients with unremarkable clinical examination and axillary ultrasound showed non-inferiority to sentinel lymph node (SLN) biopsy (SLNB). The study population consisted of 87.8% postmenopausal patients with estrogen receptor-positive, human epidermal growth factor receptor 2-negative BC. Patients with clinically node-negative breast cancer and up to two positive SLN can safely be spared axillary dissection (ALND) even in the context of mastectomy or extranodal extension. In patients enrolled in the TAXIS trial, adjuvant systemic treatment was shown to be similar with or without ALND despite the loss of staging information. After neoadjuvant chemotherapy (NACT), targeted lymph node removal with or without SLNB showed a lower false-negative rate to determine nodal pathological complete response (pCR) compared to SLNB alone. However, oncologic outcomes do not appear to differ in patients with nodal pCR determined by either one of the two concepts, according to a recently published global, retrospective, real-world study. Real-world studies generally have a lower level of evidence than RCTs, but they are feasible quickly and with a large sample size. Another global real-world study provides evidence that even patients with residual isolated tumor cells can be safely spared from ALND. In general, few indications for ALND remain. Three randomized controlled trials are ongoing for patients with clinically node-positive BC in the upfront surgery setting and residual disease after NACT. Pending the results of these trials, ALND remains indicated in these patients.

(2) Insert a reference on the system of level of evidence used as appropriate.

We thank the reviewer for this important comment. Additionally to the citation in the manuscript we have added the source of the level of evidence as “the Oxford Levels of Evidence 2” in the text of chapter 2 “Axillary surgery in the upfront surgery setting”, which now reads as follows:

Several validation studies have confirmed these results, addressed limitations of the Z0011 study, and fostered omission of ALND in these patients [19–24] (Level of Evidence [LoE] I according to the Oxford Levels of Evidence 2 [25]).